# Kalkitoxin: A Potent Suppressor of Distant Breast Cancer Metastasis

**DOI:** 10.3390/ijms24021207

**Published:** 2023-01-07

**Authors:** Saroj Kumar Shrestha, Kyung Hyun Min, Se Woong Kim, Hyoungsu Kim, William H. Gerwick, Yunjo Soh

**Affiliations:** 1Department of Dental Pharmacology, School of Dentistry, Jeonbuk National University, Jeonju 561-756, Republic of Korea; 2Laboratory of Pharmacology, School of Pharmacy, Jeonbuk National University, Jeonju 561-756, Republic of Korea; 3College of Pharmacy, Ajou University, San 5, Woncheon-dong, Youngtong-gu, Suwon 443-749, Republic of Korea; 4Center for Marine Biotechnology and Biomedicine, Scripps Institution of Oceanography and Skaggs School of Pharmacy and Pharmaceutical Sciences, University of California at San Diego, La Jolla, CA 92037, USA

**Keywords:** kalkitoxin, breast cancer, metastasis, EMT markers, an anti-cancer medicine

## Abstract

Bone metastasis resulting from advanced breast cancer causes osteolysis and increases mortality in patients. Kalkitoxin (KT), a lipopeptide toxin derived from the marine cyanobacterium *Moorena producens* (previously *Lyngbya majuscula)*, has an anti-metastatic effect on cancer cells. We verified that KT suppressed cancer cell migration and invasion in vitro and in animal models in the present study. We confirmed that KT suppressed osteoclast-soup-derived MDA-MB-231 cell invasion in vitro and induced osteolysis in a mouse model, possibly enhancing/inhibiting metastasis markers. Furthermore, KT inhibits CXCL5 and CXCR2 expression, suppressing the secondary growth of breast cancer cells on the bone, brain, and lungs. The breast-cancer-induced osteolysis in the mouse model further reveals that KT plays a protective role, judging by micro-computed tomography and immunohistochemistry. We report for the first time the novel suppressive effects of KT on cancer cell migration and invasion in vitro and on MDA-MB-231-induced bone loss in vivo. These results suggest that KT may be a potential therapeutic drug for the treatment of breast cancer metastasis.

## 1. Introduction

Breast cancer is prevalent in women, and bone is a common metastatic site of advanced breast cancer [1]. The survival rate descends from 90% in circumscribed breast cancer to metastatic breast cancer [2]. There is a relationship between metastatic breast cancer cells and bone cells, acknowledged as the “vicious cycle” [3]. Bone metastasis causes bone complications such as pathological fractures, hypercalcemia, bone pain, and spinal cord compression syndrome, causing increased morbidity and mortality in patients [4].

Epithelial-mesenchymal transition is the process of obtaining a mesenchymal phenotype by losing epithelial characteristics. Epithelial-mesenchymal transition (EMT) plays a role in increasing belligerent, invasive, and metastatic potential in cancer cells. The process of EMT favors cells undergoing phenotypic changes and molecular variations representing mesenchymal differentiation [5]. EMT leads to cancer cells losing epithelial markers, such as E-cadherin, α-catenin, and γ-catenin, and gaining mesenchymal markers fibronectin, vimentin, and N-cadherin, leading epithelial cancer cells to become more invasive and gain metastatic capabilities. EMT shows a substantial effect on drug resistance [6]. Activation of JAK2/STAT3 signaling facilitates the growth of breast cancer cells [7]. Phosphorylation of STAT3 favors breast cancer progression and activates HIF-1α [8]. Once cancer cells metastasize to the bone, different chemokine motif ligands (CXCL12, CXCR4, CXCL5, CXCR2) promote breast cancer cell colonization within the bone [9,10].

Breast cancer can produce osteoclastogenesis factors, such as PTHrP and interleukins (IL)-1, IL-6, and IL-11. These factors act on osteoblasts to increase the production of RANKL [11,12]. RANK and RANKL binding activate tumor necrosis factor receptor-associated factor 6 (TRAF6), resulting in the activation of intracellular actions, such as NF-κB; mitogen-activated kinases (MAPKs), such as JNK, ERK, and p38 signaling pathways, and nuclear factor of activated T cells c1 (NFATc1) signaling pathways. These pathways are crucial for osteoclast formation [1,12,13]. NFATc1 targets an osteoclast-specific gene such as cathepsin K, MMP9, or tartrate-resistant acid phosphate (TRAP) [13,14].

*Moorea producens* is a special kind of filamentous cyanobacteria found in marine from where kalkitoxin (KT) was isolated for the first time [15,16]. Later, kalkitoxin was extracted from cyanobacterium *Lyngbya majuscula*, an active lipopeptide found at the bottom of coral reefs. KT reduces cell division, inhibits inflammations, and was found to be ichthyotoxic [17]. Furthermore, kalkitoxin was used to observe the interaction of tetrodotoxin, and voltage-sensitive sodium channels showed substantial tumor-selective cytotoxicity that was enhanced during the period of clonogenic assays [18,19]. However, the molecular mechanism of KT in breast cancer metastasis is unknown, and the protective impact of kalkitoxin on bone destruction in vivo is not fully understood. Therefore, our main aim is to determine the effect of KT in the anti-metastatic development of breast cancer and anti-osteoclastogenesis.

## 2. Results

### 2.1. Kalkitoxin (KT) Affects Cell Viability in Breast Cancer Cell Line MDA-MB-231

To determine KT concentration for in vitro studies, we performed MTT and colony formation assays with 0, 25, 50, and 100 nM of KT. Drug concentration with cytotoxicity can affect the in vitro cell migration assay, so we treated MDA-MB-231 cells using 25, 50, and 100 nM of KT. KT is not cytotoxic at 24 and 48 h. The IC_50_ of KT is 27.64 µM for MDA-MB-231. Furthermore, there was no statistical significance even though cell viability was decreased by around 10% in KT of 50 and 100 nM at 48 h (Figure 1G). In addition, the colony-forming assay showed that KT of 25, 50, and 100 nM formed colonies of 40, 41, and 37, respectively, while without KT treatment, it formed 42 colonies (Appendix A). Taken together, KT did not affect cell proliferation below 100 nM.

### 2.2. KT Reduces the Motile Ability of MDA-MB-231

Wound healing, transwell migration, and invasion assays were performed to determine KT’s effect on cell migration. A wound healing assay was carried out within 16 h after KT treatment, whereas migration and invasion assays were carried out after 24 h of KT treatment. As a result, wound healing ability was significantly decreased in KT of 25, 50, and 100 nM compared with control (Figure 1A,B). The transwell migration assay shows that the respective concentration of KT suppresses the migration of cancer cells. The 50 and 100 nM concentrations of KT inhibit cancer cell invasion (Figure 1C–E). Our results indicate that KT reduces the motility of MDA-MB-231 breast cancer cells.

### 2.3. KT Inhibits Communication between MDA-MB-231 Cells and Active Osteoclast

Bone is a frequent site of metastases. For cancer metastasis, it has been reported that communication between tumor cells at the primary site and various cytokines are secreted from bone cells [20]. Therefore, we supposed that some factors secreted from active osteoclasts might relocate cancer cells to the bone and change tumor cell behavior. To identify whether KT could affect the correlation between the cancer cell and osteoclasts, a transwell chemotactic assay was performed on MDA-MB-231, and conditioned media from mature osteoclasts (OcCM) were applied to MDA-MB-231 cells. The transmigration of tumor cells exposed to OcCM was increased compared with traditional media. Furthermore, in OcCM with KT, the transmigrated-cell number was significantly reduced (Figure 1C,F).

### 2.4. KT Alters the Expression of Epithelial-Mesenchymal Transition Markers

EMT protein plays a significant role in cancer cell metastasis from primary to secondary tumor sites. The EMT program’s execution involves the transcriptional alteration of many genes regulating cell adhesion, mesenchymal differentiation, cell migration, and invasion [21]. Inhibiting HIF-1 activity in mice after orthotopic transplantation of triple-negative breast cancer dramatically affects primary tumor growth and metastasis to lymph nodes and lungs. HIF-1 stimulates EMT by upregulating or repressing EMT-associated transcription factors [22]. Western blot analysis showed that KT significantly enhanced claudin and E-cadherin, whereas it substantially suppressed the expression of ZEB-1, Snail, Slug, N-cadherin, β-catenin, and vimentin. In addition, KT suppressed the expression of HIF1α, MMP-2, and MMP-9, which also play a significant role in bone metabolism and metastasis of cancer (Figure 2A–K). These results show that KT shows substantial effects on EMT markers.

### 2.5. KT Suppresses the Expression of JAK2/STAT3

JAK2/STAT3 is responsible for breast cancer progression. JAK-2, STAT3, and STAT5 regulate genes that promote breast cancer cell survival, proliferation, and metastasis [8]. As shown in Figure 3A–D, protein expression of p-JAK2, p-STAT3, and p-STAT5 were markedly reduced by KT treatment. This result indicates that KT suppressed the JAK2/STAT3 pathway and prevented breast cancer cell survival and metastasis.

### 2.6. KT Inhibits the Expression of MAPK and Akt

MAPKs and Akt pathways were involved in breast cancer cell proliferation and metastasis as well as in facilitating tamoxifen resistance [23,24]. Therefore, we examined KT’s effect on the expression of MAPKs and Akt pathways. As shown in Figure 4A–E, protein expression of pERK, pJNK, p-p38, and pAkt was markedly reduced by KT treatment after 30 min. Thus, these results suggest that KT suppressed MAPKs and Akt pathways and prevented cancer cell proliferation and metastasis.

### 2.7. KT Affects the Expression of Osteoclast Factors

Cathepsin K, OPG, and NF-kB play a crucial role in osteolysis and cancer cell migration. Calcyclin-binding protein (CYBP) has been reported as a migration suppressor [25], and cathepsin K is well known as a positive regulator of bone resorption, cancer progress, and cancer cell migration. The expression of CYBP was increased by around two times compared to the control, whereas that of cathepsin K was decreased dose-independently after treatment with KT. OPG produced by the bone reduces bone degradation. However, OPG produced by cancer cells increases bone resorption. Besides the role of NF-kB in osteolysis, it also plays a role in promoting tumor cell proliferation and survival. KT inhibits the expression of both OPG and NF-kB (Figure 4A–F).

### 2.8. KT Inhibits Breast Cancer Metastasis in Mouse Modeling

Protein expression of CXCL5/CXCR2 and CXCL12/CXCR4 were decreased with KT treatment, resulting in KT inhibiting cancer cells’ secondary growth within the bone (Figure 3E–I). To evaluate whether KT could block cancer metastasis in vivo, KT was administered a day after the intracardiac injection of cancer cells, according to time schedules. The intensity of cancer signals was different in the vehicle and KT-treated groups. KT effectively inhibited cancer metastasis to secondary sites (Figure 5A). Out of eight mice in each group, the number of distant metastases was higher in the vehicle group compared to the KT-treated group. Eight of eight mice had bone metastasis in the vehicle group, while five and three mice had bone metastasis in the 2 and 10 mg/kg KT-treated groups, respectively. The intensity of cancer cells on bone was very low in KT-treated groups. Cancer cells metastasize to the brain in six out of eight in the vehicle, while KT abrogated it in KT-treated groups. The 10 mg/kg KT treatment completely inhibited breast cancer metastasis to the lungs (Figure 5B). No mice died in the 10 mg/kg KT group, while the 2 mg/kg KT group and vehicle group showed 70% and 25% survival rates, respectively (Figure 5C). There was no weight loss except for the vehicle group (Figure 5D). Immunohistochemistry analysis of the femurs of mice showed that KT substantially suppressed the expression of vimentin and β-catenin, whereas it enhanced the expression of E-cadherin. Interestingly, KT suppressed the expression of mammaglobin A in the brain of the mice dose-dependently (Figure 5E). Mammaglobin A is an attractive target for the treatment of patients with breast cancer because of its highly specific expression from breast cancers.

The µCT analysis of the femur (Figure 6A) showed the bone volume (BV)/trabecular volume (TV), bone mineral density (BMD), trabecular thickness (Tb. Th), and trabecular number (Tb. N) of the KT-treated group, which was significantly increased compared to the vehicle group and total porosity of KT-treated group and decreased compared to that of the control group (Figure 6B–F). These results indicated that KT suppressed distant cancer metastasis, indicating that KT could be an anti-cancer medicine for breast metastasis. TRAP staining of the femur shows more TRAP+ cells in the vehicle than in the sham group. Treatment of low and high KT doses significantly suppresses the number of TRAP+ cells (Figure 6G,H).

## 3. Discussion

Breast cancer frequently spreads to the bone and leads to extensive bone destruction. In the present study, we investigated whether KT had a therapeutic effect on distant metastatic cancer. Firstly, we determined a non-toxic concentration of KT to identify its impact on cell motility in the MDA-MB-231 triple-negative breast cancer cell line. According to in vitro studies, KT reduced cell migration dose-dependently; 100 nM of KT significantly reduced cell migration without cytotoxicity.

The metastatic cascade process can be explained simply in three steps: proliferation of the primary tumor, local invasion of the tumor by entrance into blood vessels (intravasation) and lymph nodes, and colonization and outgrowth of primary cancer in distant secondary organs. The close collaboration of cancerous cells with specific microenvironment elements is required in each stage [26]. Metastatic cancer usually relocates and regrows in specific organs; for example, breast cancer cells mainly grow in bones, the brain, and the lungs. This phenomenon has been explained based on the ”seed and soil” hypothesis proposed by Stephen Paget in 1889 [27,28]. The bone micro-environment is affected by cancer cells present at the primary site, thus enhancing tumor cells’ capacity to seed the bone and secrete many chemokines that control various behaviors of tumor cells to sites that support their growth [20].

Based on these reports, we speculated that osteoclasts or osteoblasts have a chemo-attractive effect on the breast cancer cell. However, according to our previous study, KT inhibited osteoclast activation. Thus, we selected osteoclast cells to verify the KT effect on communication between breast cancer cells and bone cells. Conditioned media from osteoclasts were collected and then applied to MDA-MB-231 cells. Interestingly, the transmigration and migration ability of MDA-MB-231 cells were highly induced when exposed to OcCM. However, in KT-containing OcCM, they were significantly reduced. These results suggested that unknown factors secreted from osteoclasts had a functional effect on chemotaxis and migration ability in MDA-MD-231 cells, and KT interrupted communication between breast cancer and osteoclasts.

A high level of HIF-1α at diagnosis signifies early relapse and metastasis [29], and increased expression of HIF-1 targeted genes are found in the triple-negative breast cancer subgroup [30]. HIF-1 plays a significant role in breast cancer biology, including angiogenesis, stem cell maintenance, EMT, invasion, metastasis, and obstruction of radiation and chemotherapy [31]. HIF-1 promotes EMT by controlling the transcription of E-cadherin, Snail, and ZEB-1 [32]. KT suppresses the expression of HIF-1α. JAK2/STAT3 activation promotes breast cancer cell growth, survival, and metastasis via EMT induction and chemotherapy resistance [33]. Matrix metalloproteinase (MMP) is a zinc-dependent endopeptidase that degrades many extracellular matrix (ECM) components and the bone matrix. MMP-2 and MMP-9 have corresponded positively with a higher metastasis incidence [34]. KT inhibits the expression of MMPs, suppressing bone metastasis and bone degradation. Standard features of EMT are loss of E-cadherin expression and upregulation of N-cadherin, called the “cadherin switch,” associated with increased migratory and invasive behavior of cancer cells [35]. E-cadherin, a cell-cell adhesion molecule, has been identified as a tumor suppressor protein. Loss of E-cadherin leads to tumor cell dissociation and enhanced ability to migrate, invade, and metastasize [36]. ZEB-1 is an inducer of EMT and cancer progression by downregulating E-cadherin, leading to advanced disease or metastasis [37]. KT increases the expression of E-cadherin, decreases N-cadherin, and decreases the ZEB-1, facilitating the inhibition of invasion from the epithelium to the blood. The first family member identified, claudin 1, forms the tight junction strand. The PDZ domain of claudin 1 interacts with zona occludins (ZO1 and ZO2), which connect to several signaling pathways [38]. The overexpression of claudin 1 alone induced apoptosis in MDA-MB-231 cells [39]. KT significantly increases claudin expression, facilitating lower recurrence status and more prolonged disease-free survival in breast cancer; however, overexpression of claudin 1 increases cell invasion in colon cancer [40].

Vimentin contributes to the aggressive phenotype in invasive breast cancer and is considered a canonical marker of EMT, which plays a role in tumor invasion and progression [41]. β-catenin is a cytoplasmic plaque protein that plays a vital role in EMT [42]. In cells undergoing EMT, it is located either in the nucleus or cytoplasm of cells. Localization of β-catenin to the nucleus stimulates the transcription of genes that induce EMT [43]. KT inhibits the expression of both vimentin and β-catenin, suppressing the invasion of cancer cells. The Snail family includes three members as Snail, Slug, and Smuc. Snail is widely known as the repressor of E-cadherin and activator of nuclear ERK2 during EMT [44]. Slug enhances the colony formation size and number [45] and plays a role in the homing/colonization of cancer cells to bone [46]. The CXCL5/CXCR2 axis stimulates bone colonization, which is also a step for cancer cell stability in bone [9]. KT inhibits the Slug and CXCL5/CXCR2 axis, inhibiting colony formation within the bone after metastasis.

KT regulated the expression of the calcyclin-binding protein (CYBP) and cathepsin K. CYBP has been reported as a negative regulator of malignant behavior in gastric cancer [25] and metastasis in colon cancer [47]. Cathepsin K is a protease responsible for bone resorption. Recently developed potent cathepsin K inhibitors act as useful anti-resorptive agents to treat osteoporosis [48]. In general, bone loss is caused by the over-activation of osteoclasts that breaks down bone tissue. Tumor cells within the bone promote osteolysis, creating a favorable micro-environment as the bone matrix is abundant in growth factors. Tumor cells in bone secrete pre-osteoclast maturing factors, such as parathyroid hormone-related protein, IL-11, and TNF-ɑ, which stimulate osteoblasts to increase RANKL and decrease osteoprotegerin (OPG) production, consequentially leading to bone destruction [49]. OPG secreted from bone marrow acts as a negative regulator of bone metabolism. Still, OPG secreted from breast cancer cells enhances the tumor-promoting effect on primary breast tumors by blocking TRAIL action and through a direct impact on tumor cells. KT inhibits the OPG, which promotes the growth of endothelial cells and tubule formation and promotes metastatic tumors at sites outside of bone [50]. RANKL, a ligand of RANK’s receptor, has been considered the central molecule for bone destruction. The binding of RANKL to RANK stimulates the maturation and activation of osteoclasts. OPG, a decoy receptor of RANKL, blocks the binding of RANKL to RANK, leading to the inhibition of osteoclast differentiation and activation. The RANKL/RANK/OPG system imbalance is common in several tumors such as breast cancer, prostate cancer, malignant bone tumors, and squamous cell carcinoma [51,52].

Chemotherapy is a standard systemic treatment for bone metastasis, and many drugs for breast cancer treatment mainly target bone. The bisphosphonate (BP) group of anti-cancer drugs is widely used in general medicine. BPs inhibit osteoclast development and migration, increase the production of OPG by osteoblasts, and promote osteoclast apoptosis [53]. RANKL is considered the central molecule for bone resorption. Anti-RANKL drugs inhibit osteoclast differentiation by masking the RANKL receptor. In our previous studies, KT inhibited osteoclast activation and regenerated bone destruction by inflammation in the osteoporosis mouse model.

KT was initially isolated from *Lyngbya majuscula* to collect new molecules for testing as anti-tumor or anti-fungal agents [54]. However, until now, KT’s effect on cancer has only been reported in 2015. This paper explained that KT inhibits HIF-1 activation by suppressing mitochondrial oxygen consumption at the electron transport chain and targets tumor angiogenesis by blocking the induction of angiogenic factors (i.e., VEGF) in tumor cells [16].

Based on our in vitro studies, KT verifies the therapeutic effect on metastatic cancer and bone loss caused by cancer in the bone. Metastatic mouse modeling was established through the cardiac injection of MDA-MB-231_Luc2 cells. Firstly, we examined whether KT could inhibit cancer metastasis through an in vivo mouse model. Still, the intensity of cancer signals in secondary organs showed significant differences between the vehicle and two KT-treated groups. We thus tested whether KT was adequate to reduce metastases-secondary cancer. KT was administered orally from the day after the intracardiac injection of MDA-MB-231 cells to five weeks after cancer implantation. A high dose of KT completely inhibits breast cancer metastasis to the brain and lungs. KT may be a promising agent for the treatment of metastasis to the brain and lungs besides bones. Our results showed that the treatment of KT suppressed distant cancer metastases.

Furthermore, micro-CT analysis results showed that bone loss caused by metastatic cancer decreased in the KT-treated group compared to the vehicle. Moreover, the mice’s physical conditions, such as walking, playing, and body weight were similar to normal mice in the KT group. Our previous studies identified that KT inhibited osteoclast activation and repaired inflammatory bone loss [55]. Therefore, these results may indicate KT’s function in bone regeneration by suppressing osteoclast differentiation. Indeed, it was challenging to determine KT’s effect on the bone with the present results, but we consider these findings significant, and further studies are needed.

## 4. Materials and Methods

### 4.1. Reagents

Fetal bovine serum (FBS), α-modified essential medium (α-MEM), and penicillin were purchased from Gibco (Gaithersburg, MD, USA). Parental MDA-MB-231 cells were purchased from a Korean Cell Line Bank (South Korea). RAW264.7 macrophage cells were purchased from ATCC (Manassas, VA, USA). RANKL was obtained from PeproTech (Rocky Hill, NJ, USA. pGL4.50 [luc2/CMV/] vector was purchased from Promega (Madison, WI, USA). The antibiotic hygromycin was purchased from Gibco BRL (Gaithersburg, MD, USA). Primary antibodies were purchased from Thermo Fisher Scientific (Waltham, MA, USA), Santa Cruz Biotechnology (Santa Cruz, CA, USA), and Abcam (Cambridge, MA, USA).

### 4.2. Cell lines and Conditioned Medium

Human MDA-MB-231 breast cancer cells and MDA-MB-231-luc2 cells were cultured in a standard DMEM supplemented with 10% fetal bovine serum (FBS), penicillin (100 U/mL), and streptomycin (100 U/mL) and maintained at 37 °C in 5% CO_2_.

### 4.3. Cell Viability

The cytotoxicity of KT was examined with the MTT assay. A total of 4 × 10^3^ of MDA-MB 231 cells/well were seeded on 96 well plates separately and treated with various KT doses for 24 h and 48 h. KT untreated cells were taken as control. Then 0.5 mg/mL of MTT was added to cells for 2 h at 37 °C, and cells were dissolved with 200 μL dimethyl sulfoxide (DMSO) and quantified with a spectrophotometer at 570 nm.

### 4.4. Wound–Healing Assay/Transwell Migration and Invasion Assay

MBA-MB-231 cells were maintained until nearly 100% confluence, and then a linear scratch was made using a 200 μL pipette tip. After washing twice with phosphate-buffered saline (PBS), the cells were treated with KT of 0, 25, 50, and 100 nM and then incubated for 16 h, and images were taken at 10× magnification.

According to the manufacturer’s protocol, a transwell chamber with 24-well, 8.0 μM pore membranes (Corning USA) was used. A total of 1 × 10^5^ cells/well were seeded in the upper chamber with 100 μL serum-free medium, and 600 μL of complete medium was added to the lower chamber as a chemoattractant with or without KT. After 24 h of incubation, cells on the upper surface were removed with a cotton swab, and cells on the lower surface were fixed with 4% paraformaldehyde and stained with 0.1% crystal violet solution. Cells were photographed with an inverted fluorescence microscope.

For the osteoclast soup migration assay, 5 × 10^4^ cells per well were seeded in the upper chamber in 100 μL serum-free medium, and 600 uL of OcCM was added to the lower chamber chemoattractant with different concentrations of KT for 24 h.

For the transwell invasion assay, 2 × 10^5^ cells were seeded, and the assay was carried out in a similar manner to the transwell migration assay described above, except that 100 μL of 1:8 DMEM/dilute Matrigel (BD, USA) was added to each well at 37 °C for 2 h before the cells were seeded on the membrane.

### 4.5. Preparation of Total Protein and Conditioned Media

Cells were lysed in a protein extraction buffer (50 mM Tris (pH 8.0), 5 mM EDTA, 150 mM sodium chloride, 0.5% deoxycholic acid, 0.1% SDS, 1% NP-40, 1 mM PMSF, and 1 mg/mL protease inhibitor cocktail) to prepare total protein. Cells were then centrifuged, and the supernatant was retained. The concentrations of all protein samples were determined using a Bio-Rad Protein Assay Kit (Bio-Rad, Hercules, CA, USA).

RAW264.7 macrophage cells were exposed to RANKL for differentiation into osteoclast-like cells to obtain conditioned media from osteoclasts (OcCM). RAW264.7 cells were cultured with growth media until 80% confluence and then exposed to differentiation media containing 2% FBS with 50 ng/mL RANKL for five days. The media were then replaced with fresh media with 2% FBS. After 24 h, conditioned media were clarified by centrifugation and stored at −80 °C until use.

### 4.6. Western Blot Analysis

MDA-MB-231 cells were lysed in a lysis buffer, 10% SDS-PAGE was used to separate protein, and these proteins were moved to a polyvinylidene difluoride membrane using a transferred buffer. Nonfat skimmed milk (5%) was prepared, and membranes were blocked for 60 min at room temperature. Blocked membranes were treated with primary rabbit anti-p-ERK, anti-p-p38, anti-p-JNK, anti-ERK, anti-JNK, anti-p38, anti-NFATc1, anti-cathepsin K, and anti-β-actin antibodies, which were diluted at 1:500 to 1:1000 ratios and incubated at 4 °C for 15 h. Secondary conjugated anti-rabbit or anti-mouse antibodies were diluted at a 1:3000 to 1:5000 ratio in 5% skimmed milk for 2 h at room temperature. Then protein expression was developed with an ECL detection kit (Bio-Rad, Hercules, CA, USA).

### 4.7. Establishment of Luciferase-Expressing Stable Cells

MDA-MB-231 cells were transfected with pGL4.50 (Luc2/CMV/Hygro) vector using LipofectamineTM 2000 (Invitrogen, Carlsbad, CA, USA) according to the manufacturer’s instructions. Positive clones of MDA-MB-231 cells were selected after treatment with 300 µg/mL hygromycin for one week, and the IVIS system confirmed the luciferase activity of cells. These cell lines were designated as “MDA-MB-231_Luc2”.

### 4.8. Tumor Implantation and Drug Treatment

The ethics committee for animal handling at Jeonbuk National University, South Korea, approved the study. All mice experiments were performed in accordance with the local institutional animal care and use committee (IACUC) guidelines. Four-week-old female athymic nude mice were used to establish a breast cancer metastasis model. All mice were obtained from NARA Biotech (Tae-jeon, South Korea). Mice were divided into 4 groups of 8 mice: sham, vehicle, low-dose (2 mg/kg), and high-dose (10 mg/kg). For intracardiac injection, vehicle and dose groups of mice were anesthetized with 250 mg/kg of Avertin (2, 2, 2-Tribromoethanol), then 10^5^ MDA-MB-231-Luc2 cells were suspended in 100 µL of phosphate-buffered saline and injected into the left ventricle of the heart. Mice of the standard control group were injected intracardially with PBS instead of cancer cells. The cancer development and metastasis in the mice were monitored once at five weeks through the IVIS system. The mice in KT-treated groups were orally administered KT once per two days for five weeks. At the experimental endpoint, all mice were euthanized. Bone fragments and brain were excised, fixed in 4% paraformaldehyde overnight, and stored at −80 °C until further analysis.

### 4.9. In Vivo Bioluminescent Imaging (BMI)

After cancer implantation to the heart, BLI of femurs was measured at five weeks, and signal intensity of KT-treated groups was compared with the vehicle group. Bioluminescent imaging was conducted with an in vivo optical imaging system (IVIS 200; Xenogen Corp.). During the imaging procedure, all mice were anesthetized with isoflurane gas (2% isoflurane in oxygen, 1 L/min). Five minutes after the intraperitoneal injection of 100 μL of D-luciferin solution (15 mg/mL), all mice acquired a bioluminescence image. Living Image software (Xenogen Corp) analyzed signals by depicting a circular ROI around each bioluminescent source. The results are reported as the maximum photon flux within an ROI in photons per second (photon·s^−1^·cm^−2^·sr^−1^).

### 4.10. Assessment for Osteolytic Bone Metastases

Both femurs of mice were analyzed using a high-performance in vivo micro-CT Skyscan 1076 (Skyscan, Kontich, Belgium), with an image field at a pixel size of 13 µM. The distal femoral metaphysis was analyzed from a region that was 1.0 mM below the growth plate and 1.5 mM in length. For quantitative analysis, the software CTAn (Skyscan) was used to obtain the following parameters within the region of interest (ROI): bone volume/tissue volume (BV/TV), trabecular thickness (Tb. Th), trabecular separation (Tb. Sp), trabecular number (Tb. N), and trabecular bone mineral density (BMD) [49]. The cortical BMD was analyzed in the middle of the diaphysis of the femur.

### 4.11. Immunohistochemistry

Femurs were fixed in 10% neutral-buffered formalin, decalcified with 10% EDTA for three weeks, and then embedded in paraffin. Paraffin-embedded femurs were cut at 4 μM, deparaffinized with xylene, and rehydrated with alcohol for further hematoxylin-eosin or 3,3′-diaminobenzidine peroxidase (DAB) immunohistochemical staining. Proteolytic digestion and peroxidase blocking of femur slides was carried out, and slides were incubated overnight with primary antibodies against E-cadherin, vimentin, and β-catenin dilution 1:100 at 4 °C.

### 4.12. Statistical Analysis

Experiments were performed thrice and analyzed by one–way ANOVA with Tukey’s multiple comparisons test. GraphPad Prism software was used to analyze all statistical tests (GraphPad Software Inc., La Jolla, CA, USA), and the data are expressed as means ± S.D. Values of *p* < 0.05 were appraised as statistically significant.

## 5. Conclusions

Kalkitoxin inhibits neoplastic cell metastasis and crosstalk between bone, brain, lungs, and cancer cells. KT inhibits wound healing, and the transwell migration assay indicates the inhibition of local migration of MDA-MB-231 cells within the primary site. Furthermore, KT inhibits the invasion of cancer cells, suppressing the invasion of the cancer cells to the blood vessel from a primary site to a blood vessel and blood vessels to the secondary sites. KT also alters the expression of protein markers to suppress epithelial–mesenchymal transition. E-cadherin, HIF-1α, and vimentin are major proteins for EMT, migration, and invasion, and KT strongly alters the expression of these proteins. It also delays secondary tumor growth and progression by inhibiting JAK2/STAT3 and MAPK pathways. KT alleviates the invasion of cancer cells from blood vessels to bone by inhibiting CXCL5/CXCR2 and CXCL12/CXCR4. In bone, the “vicious cycle” is disturbed by the inhibition of CtsK, OPG, and p-NFκB by the action of KT preventing bone loss caused by metastatic breast cancer. KT shows an anti-metastatic effect on MDA-MB-231 in vitro as well as in vivo, so it has the potential as an anti-cancer medicine to alleviate severe pain in patients with metastatic cancer.

## Figures and Tables

**Figure 1 ijms-24-01207-f001:**
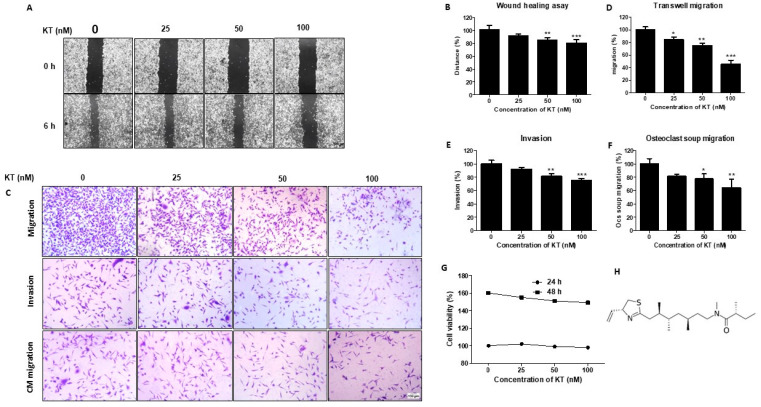
Kalkitoxin suppresses the migration of MDA-MB-231 cells. Cells were cultured with DMEM high glucose (growth media) with or without 25, 50, and 100 nM concentrations of KT. (**A**) Wound healing assay (Scale bar, 100 µm) (**C**), transwell migration, transwell invasion assay, and CM migration were observed (Scale bar, 100 µm). (**B**,**D**–**F**) Quantitative analysis of the assays with or without KT. (**G**) Effect of different concentrations of KT on MDA-MB-231 cells at 24, 48, and 72 h, observed by MTT assay. (**H**) Molecular structure of kalkitoxin. Results are presented as means ± S.D. (*n* = 3). * *p* < 0.05, ** *p* < 0.01, and *** *p* < 0.001 indicate significant differences between KT-treated groups and control.

**Figure 2 ijms-24-01207-f002:**
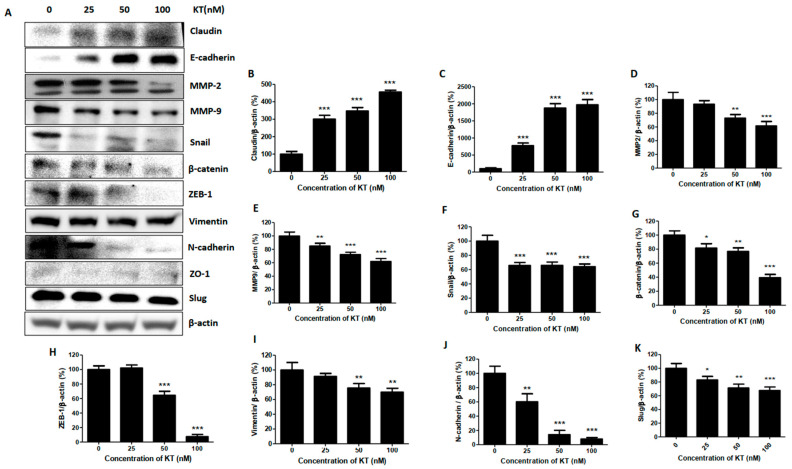
KT suppresses breast cancer epithelial-mesenchymal markers. Cells were cultured with DMEM high glucose (growth media) with or without 25, 50, and 100 nM concentrations of KT. (**A**) Western blot analysis was performed to determine the expression of breast cancer EMT marker proteins. (**B**–**K**) Quantitative analysis of the expression of EMT marker proteins. Results are presented as means ± S.D. (*n* = 3). * *p* < 0.05, ** *p* < 0.01, and *** *p* < 0.001 indicate significant differences between KT-treated groups and control.

**Figure 3 ijms-24-01207-f003:**
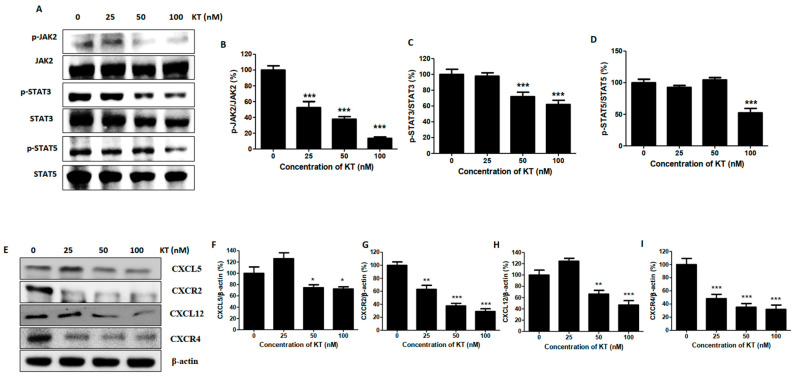
KT suppresses JAK2/STAT3 pathways and abrogates secondary cancer cell growth on bone. C. (**A**) Western blot analysis was performed to determine the expression of JAK2/STAT3 proteins. (**B**–**D**) Quantitative analysis of related protein expression. (**E**) Protein expression of secondary cancer cell growth factors on bone. (**F**–**I**) Quantitative analysis of the secondary growth stimulator of cancer cells on bone proteins. Results are presented as means ± S.D. (*n* = 5). * *p* < 0.05, ** *p* < 0.01, and *** *p* < 0.001 indicate significant differences between KT-treated groups and control.

**Figure 4 ijms-24-01207-f004:**
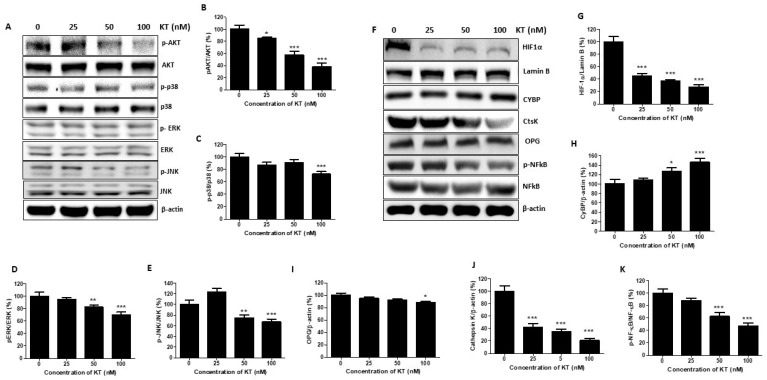
KT suppresses the MAPK pathway and osteoclastogenic protein expression. MDA-MB-231 cells were cultured in DMEM high glucose (growth media) with or without 25, 50, and 100 nM of KT for 30 min for the MAPK pathway and 24 h for osteoclastogenic proteins. (**A**) The total protein was isolated to perform Western blotting using antibodies p-AKT, p-JNK, p-ERK, and p-JNK. (**B**–**E**) Quantitative analysis of MAPK pathway proteins. (**F**) Western blot analysis was performed to determine the expression of osteoclastogenic-related proteins. (**G**–**K**) Quantitative analysis of osteoclastogenic-related protein expression. Results are presented as means ± S.D. (*n* = 3). * *p* < 0.05, ** *p* < 0.01, and *** *p* < 0.001 indicate significant differences between KT-treated groups and control.

**Figure 5 ijms-24-01207-f005:**
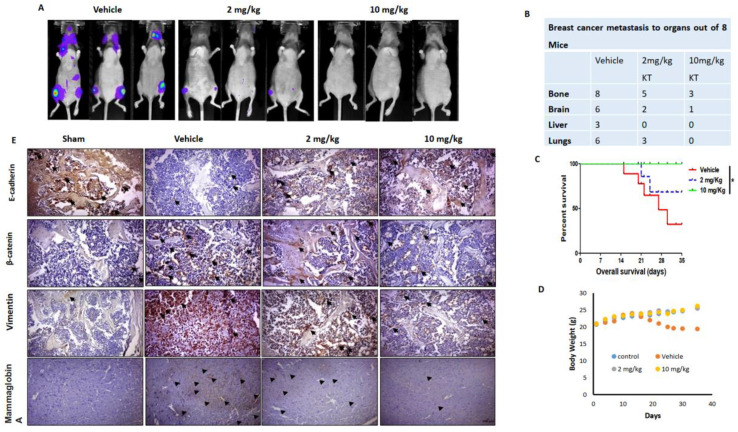
KT inhibits distant breast cancer metastasis in vivo. (**A**) MDA-MB-231 cells were intracardially injected into the mice. After five weeks with or without treatment of KT, each group of mice had bioluminescence after five weeks. (**B**) Several distant metastases in mice of each group. (**C**) The Mantel-Cox log-rank test was used to determine percentage survival. (**D**) Average mouse weights during the course of the study were measured. (**E**) Femurs were fixed, decalcified, embedded, sectioned, and subjected to immunohistochemistry for EMT marker proteins such as E-cadherin, β-catenin, and vimentin in the femur section (Scale bar, 100 µm). Arrow indicates the dense staining of the biomarker. Immunohistochemistry for mammaglobin A protein in the brain section was performed. Results are presented as means ± S.D. (*n* = 8). * *p* < 0.05 indicates significant differences between KT-treated groups and vehicle.

**Figure 6 ijms-24-01207-f006:**
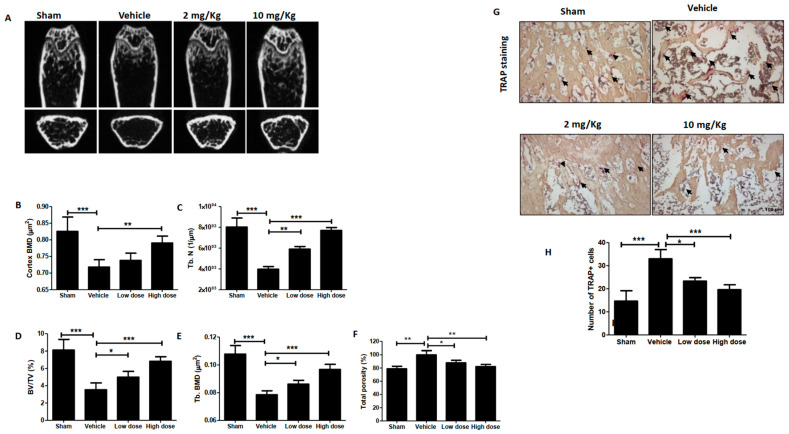
KT inhibits osteolysis in vivo. (**A**) MDA-MB-231 cells were intracardially injected into the mice. After five weeks with or without treatment of KT, representative micro-CT images of each group’s mice were shown. (**B**–**F**) BV/TV, trabecular separation, cortex BMD, trabecular number, trabecular thickness, trabecular BMD, and total porosity were determined using micro-CT data analysis. (**G**) Femurs were fixed, decalcified, embedded, and sectioned. TRAP staining of mice and osteoclast numbers in femurs were counted (Scale bar, 100 µm). Arrows indicate mature osteoclasts. (**H**) Quantitative analysis of TRAP-positive osteoclast numbers displayed as a histogram. Results are presented as means ± S.D. (*n* = 8). * *p* < 0.05, ** *p* < 0.01, and *** *p* < 0.001 indicate significant differences between KT-treated groups and control.

## Data Availability

The data presented in this study are available on request from the corresponding author. The data are not publicly available due to privacy.

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
