# Peer review of "Kalkitoxin: A Potent Suppressor of Distant Breast Cancer Metastasis"

_ijms, 2023, doi:10.3390/ijms24021207_

Round 1

Reviewer 1 Report

Kalkitoxin: a potent suppresser of distant breast cancer metastasis

This is a very interesting study, that uses calcitoxin (KT), a lipopeptide toxin derived from the marine cyanobacteria Moorena Producens (formerly Lyngbya majuscule) in a breast cancer model. The authors show that KT suppressed the migration and invasion of cancer cells in in vitro models and animals. It was shown that KT suppressed the invasion of MDA-MB-231 cells in vitro and induced osteolysis in a mouse model, possibly increasing/inhibiting markers of metastasis. Furthermore, KT also inhibited the expression of CXCL5, and CXCR2, suppressing metastasis in bones, brain, liver and lungs. Finally, they demonstrated a protective role by computed microtomography and immunohistochemistry, suggesting that KT may be a potential therapeutic drug for the treatment of breast cancer metastases.

The article is well-organized and contains all the components necessary. The sections are well-developed, and the state of the art is current and complete, supporting the author's questions. The methodology is clearly explained, and the showed results are discussed properly. The text is easy to understand.

However, there are a few suggestions to improve it.

Methods:

Line 344: 4.5. Cell viability: Let clear which control was used.

Comment: line 404: The doses ( 2 and 10 mg/kg) were defined based on what?

Line 350 :  Wound–healing assay. Was any cell proliferation inhibitor used? Why only watched for 6 hours? Was it observed longer?

Line 446: conclusion. The conclusions were written as if they were "hallmarks". There are many separate phrases. Please consider writing the conclusions in a continuous text.

Line: 4.11. Tumor implantation and drug treatment

Results:

In general, the figures are very well done.

Figure 1: the magnification bar is missing in figure 1A

Line 75: The IC50 of KT is 27.64 µM. Is the unit µM or nM? I suggest the use of the same unit.

Line - 78 -  In addition, the colony-forming assay showed that KT did not affect cell proliferation below 100 nM (Supp Fig. 1).

Comment: Could you consider, in addition to the number of colonies, the effect of different KT concentrations on the size of colonies formed?

Line 87: Comment: Considering if the IC50 is 27.64µM, it is assumed that any higher concentration may act on mobility, as well as kill the cells. The same consideration must be considered for invasion and migration.

But if the IC50 is 27.64nM, please disregard the comment.

Line 96: If possible, I suggest comparing the migration with and without OcCM

Figure 5: Immunohistochemistry: it is necessary to better position the arrowheads to effectively show the markings, especially in figure 5.

Discussion:

line 310: Based on our in vitro studies and some references…

Comment: These references must be cited

Line  325: Our previous studies (please, cite it)

 And, some words should be in italics (in vitro, in vivo).

Author Response

Dear, Mr. Petar Djurdjic                                                    2022-12-22

Editor

  Enclosed, please find our revised manuscript (ijms-2077414) entitled “Kalkitoxin: a potent suppresser of distant breast cancer metastasis,” which has been reviewed for publication in the international journal of molecular sciences.

     We are very thankful for your valuable comments. We have tried our best to complete what reviewers have asked to do and have tried to answer point-wise. All parts changed over the revised manuscript were tracked. We hope that this revision will satisfy the requirements and suggestions derived from the reviewers.

             We changed figures 1, 4, and 6 and added one piece of data in the supplementary figure, as suggested by the referees. MDA-MB-231 cells are metastasized to the liver, but other cell lines are suitable for studying cancer metastasis to the liver. So we decided to remove the data and sentences related to the liver in our manuscript. And I will request you to check the reference format because I add some references at the referee's suggestion, and later I cannot make it as in the previous reference format.

Thank you very much for your consideration.

Best regards,

Yunjo Soh, PhD

Reviewer1

Kalkitoxin: a potent suppresser of distant breast cancer metastasis

This is a very interesting study, that uses calcitoxin (KT), a lipopeptide toxin derived from the marine cyanobacteria Moorena Producens (formerly Lyngbya majuscule) in a breast cancer model. The authors show that KT suppressed the migration and invasion of cancer cells in in vitro models and animals. It was shown that KT suppressed the invasion of MDA-MB-231 cells in vitro and induced osteolysis in a mouse model, possibly increasing/inhibiting markers of metastasis. Furthermore, KT also inhibited the expression of CXCL5, and CXCR2, suppressing metastasis in bones, brain, liver and lungs. Finally, they demonstrated a protective role by computed microtomography and immunohistochemistry, suggesting that KT may be a potential therapeutic drug for the treatment of breast cancer metastases.

The article is well-organized and contains all the components necessary. The sections are well-developed, and the state of the art is current and complete, supporting the author's questions. The methodology is clearly explained, and the showed results are discussed properly. The text is easy to understand.

However, there are a few suggestions to improve it.

Methods:

  1. Line 344: 4.5. Cell viability: Let clear which control was used.
  • Author response: Thank you. KT untreated cells were taken as control. We have added this sentence in line 347.

  1. Comment: line 404: The doses (2 and 10 mg/kg) were defined based on what?
  • Author response: Thank you for the interesting question. We have already experimented with safe doses of Kalkitoxin on mice. We have found that LD50 of KT on mice is 100 mg/kg. Up to 60 mg/kg of KT treatment, no mice death occurred. So, 2 and 10 mg/kg of KT doses are safe.
  • In another experiment on the effect of KT on bone, we used 5 mg/kg of KT. This dose is effective in bone, so we choose 2 and 10 mg/kg of KT to know how these doses affect the distant metastasis of breast cancer.

  1. Line 350 :  Wound–healing assay. Was any cell proliferation inhibitor used? Why only watched for 6 hours? Was it observed longer?
  • Author response: Thank you very much. There is not any cell proliferation inhibitor was used.
  • MDA-MB-231 is a highly aggressive cell line. It will migrate very fast in comparison to other cancer cells. We found significant cancer cell migration within 16 hours, so we used data of 16 hours. We apologize for the mistake of 6 h instead of 16 h. This mistake is corrected in the article.
  • Yes, we can observe it longer than we did. We also check cell migration in 16 and 24 h (Figures are not shown because significant migration occurs within 16 h).

  1. Line 446: conclusion. The conclusions were written as if they were "hallmarks". There are many separate phrases. Please consider writing the conclusions in a continuous text.
  • Author response: The conclusion is edited as per the referee's suggestion.

  1. Line: 4.11. Tumor implantation and drug treatment
  • Author response: I do not get what the referee wants to mention in this comment.

Results:

In general, the figures are very well done.

  1. Figure 1: the magnification bar is missing in figure 1A.
  • Author response: Thank you. We have added a magnification bar in Figure 1A.

  1. Line 75: The IC50 of KT is 27.64 µM. Is the unit µM or nM? I suggest the use of the same unit.
  • Author response: Thank you very much for suggesting clearing up confusion on this line. The unit is µM. We think writing 27,640 nm is not good as 27.64 µM, and we have already mentioned in line 75 that KT does not show any cytotoxicity at 100 nM, so there is no chance of IC50 around 27.64 nM.

  1. Line - 78 -  In addition, the colony-forming assay showed that KT did not affect cell proliferation below 100 nM (Supp Fig. 1).

Comment: Could you consider, in addition to the number of colonies, the effect of different KT concentrations on the size of colonies formed?

  • Author response: Thank you. We have added the number of colonies in the article per your suggestion in line 78.

  1. Line 87: Comment: Considering if the IC50 is 27.64µM, it is assumed that any higher concentration may act on mobility, as well as kill the cells. The same consideration must be considered for invasion and migration.

But if the IC50 is 27.64nM, please disregard the comment.

  • Author response: Thank you. The IC50 is 27.64 µM. Yes, the higher the concentration of KT, the more effective against migration and invasion and more the number of cell deaths. Our target is not to kill the cancer cell but to inhibit the metastasis of cancer cells from the breast to other parts of the body like bone, brain, and lungs. KT of 50 and 100 nM are effective against migration and invasion, so we do not use a higher concentration of KT than 100 nM.

  1. Line 96: If possible, I suggest comparing the migration with and without OcCM
  • Author response: Thank you for your exciting suggestion. It is tough to compare migration with and without OcCM because we seeded different concentrations of cells, like 1 × 105 in the migration assay without OcCM and 5 × 104 in the migration assay with OcCM. But we can conclude from Figure 1C that with the lower number of cell seeding, there is a high number of cell migration in OcCM media.

  1. Figure 5: Immunohistochemistry: it is necessary to better position the arrowheads to effectively show the markings, especially in figure 5.
  • Author response: Thank you very much for the suggestion. The staining is apparent, as shown in figure 5. The arrow is shown on the site where the staining is high compared to the peripheral area.

Discussion:

  1. line 310: Based on our in vitro studies and some references…

Comment: These references must be cited

  • Author response: Thank you for the suggestion. The “Some references” phrase is removed from the article because there has been no article on the effect of KT on breast cancer until now.

  1. Line  325: Our previous studies (please, cite it)
  • Author response: Thank you. I have cited the reference as “Li, L., et al., Kalkitoxin reduces osteoclast formation and resorption and protects against inflammatory bone loss. International journal of molecular sciences, 2021. 22(5): p. 2303”.

  1. And, some words should be in italics (in vitro, in vivo).
  • Author response: Thank you for it. I have checked it and corrected it.

Reviewer 2 Report

The manuscript titled as “Kalkitoxin: a potent suppresser of distant breast cancer metastasis” elucidated the molecular mechanism of Kalkitoxin against breast cancer metastasis. This research is interesting but there are several important issues that should be addressed before the manuscript is ready for publication.

 Major issues:

1. Line 74-75. The authors found that KT didn’t exhibit cytotoxicity in the concentration range of 25-100 nM, and the IC50 value was 27.64 μM. But the authors didn’t provide relevant data to support this conclusion?

2. The authors use means ± SD in the figure caption of Fig. 1, but in the method section, the authors use means ± SEM. Please unify the mathematical and statistical methods throughout the text and make changes accordingly.

3. Line 127-128: JAK2/STAT3 is related to the occurrence and progression of breast cancer. Please cite relevant references to confirm this conclusion.

4. Line 139-140: MAPKs and Akt pathways were involved in breast cancer cell proliferation. Please cite relevant references to confirm this conclusion.

5. Fig. 4A. The JNK and p-JNK theoretically contain both p46 and p54 bands, but Fig. 4A in the manuscript showed only one band. Please explain the results and provide the original blots.

6. According to the research result, the author should summarize and give a clear mechanism as a potent suppresser of distant breast cancer metastasis.

7. The language needs to be checked and polished.

Minor issues:

1. Line 73: the usage of “Because” in Line 73 is not appropriate here.

2. Line 141-143: the relevant contents in line 141-143 could not be corresponded to Fig. 2A-E.

3. Fig. 5E. Please add the scale in immunohistochemistry images in Fig. 5E.

4. The author should provide the relevant structure of the Kalkitoxin in this manuscript.

Author Response

Dear, Mr. Petar Djurdjic                                                    2022-12-22

Editor

  Enclosed, please find our revised manuscript (ijms-2077414) entitled “Kalkitoxin: a potent suppresser of distant breast cancer metastasis,” which has been reviewed for publication in the international journal of molecular sciences.

     We are very thankful for your valuable comments. We have tried our best to complete what reviewers have asked to do and have tried to answer point-wise. All parts changed over the revised manuscript were tracked. We hope that this revision will satisfy the requirements and suggestions derived from the reviewers.

             We changed figures 1, 4, and 6 and added one piece of data in the supplementary figure, as suggested by the referees. MDA-MB-231 cells are metastasized to the liver, but other cell lines are suitable for studying cancer metastasis to the liver. So we decided to remove the data and sentences related to the liver in our manuscript. And I will request you to check the reference format because I add some references at the referee's suggestion, and later I cannot make it as in the previous reference format.

Thank you very much for your consideration.

Best regards,

Yunjo Soh, PhD

Reviewer2

Comments and Suggestions for Authors

The manuscript titled as “Kalkitoxin: a potent suppresser of distant breast cancer metastasis” elucidated the molecular mechanism of Kalkitoxin against breast cancer metastasis. This research is interesting but there are several important issues that should be addressed before the manuscript is ready for publication.

 Major issues:

  1. Line 74-75. The authors found that KT didn’t exhibit cytotoxicity in the concentration range of 25-100 nM, and the IC50value was 27.64 μM. But the authors didn’t provide relevant data to support this conclusion?
  • Author response: Thank you very much. We have attached supported data in supplementary figure 1C.
  1. The authors use means ± SD in the figure caption of Fig. 1, but in the method section, the authors use means ± SEM. Please unify the mathematical and statistical methods throughout the text and make changes accordingly.
  • Author response: Thank you. I apologize for this mistake. All data are presented as mean ± SD. We replaced SEM with SD in 4.14.
  1. Line 127-128: JAK2/STAT3 is related to the occurrence and progression of breast cancer. Please cite relevant references to confirm this conclusion.
  • Author response: Thank you. We have cited the reference as, “a, J.-h., L. Qin, and X. Li, Role of STAT3 signaling pathway in breast cancer. Cell Communication and Signaling, 2020. 18(1): p. 1-13”.
  1. Line 139-140: MAPKs and Akt pathways were involved in breast cancer cell proliferation. Please cite relevant references to confirm this conclusion.
  • Author response: Thank you very much. We have cited the reference as “Ortega, M.A., et al., Signal transduction pathways in breast cancer: the important role of PI3K/Akt/mTOR. Journal of oncology, 2020. 2020” and Vafeiadou, V., D. Hany, and D. Picard, Hyperactivation of MAPK induces tamoxifen resistance in SPRED2-deficient ERα-positive breast cancer. Cancers, 2022. 14(4): p. 954”.
  1. Fig. 4A. The JNK and p-JNK theoretically contain both p46 and p54 bands, but Fig. 4A in the manuscript showed only one band. Please explain the results and provide the original blots.
  • Author response: Thank you very much for it. Yes, JNK and p-JNNK have both bands. I selected only one band before. In the revised manuscript, I have added both bands to Figure 4A and provided the original blots too.
  1. According to the research result, the author should summarize and give a clear mechanism as a potent suppressor of distant breast cancer metastasis.
  • Author response: Thank you very much. We have written a precise mechanism in the conclusion section.
  1. The language needs to be checked and polished.
  • Author response: Thank you very much. The article was checked by a native English writer before. And we checked English later too.

Minor issues:

  1. Line 73: the usage of “Because” in Line 73 is not appropriate here.
  • Author response: Thank you very much. “Because” is removed in this sentence.

  1. Line 141-143: the relevant contents in line 141-143 could not correspond to Fig. 2A-E.
  • Author response: Thank you very much. We have corrected it as Fig. 4A-E.

  1. 5E. Please add the scale in immunohistochemistry images in Fig. 5E.
  • Author response: Thank you very much. We have already added the scale bar in Fig 5E in the bottom right corner.

  1. The author should provide the relevant structure of the Kalkitoxin in this manuscript.
  • Author response: Thank you. We have added the kalkitoxin structure in Figure 1H.

Round 2

Reviewer 2 Report

In the revised manuscript and supplementary,

1. I didn't found 1c  in Figure 1 of supplementary.

2. In Figure 4A, the bands of JNK and p-JNK are not changed. 

3. I didn't found 1H  in Figure 1.

Author Response

Dear Reviewer

      We are very thankful for your valuable comments. We have tried our best to complete what reviewers have asked to do and have tried to answer point-wise. All parts changed over the revised manuscript were tracked. We hope that this revision will satisfy the requirements and suggestions derived from the reviewers.

             We changed figures 1, 4, and 6 on the revised manuscript and added one piece of data in the supplementary figure, as suggested by the referees. I request you add the supplementary file in the appropriate way.

Thank you very much for your consideration.

Sincerely yours

Yunjo Soh, Ph.D.

Laboratory of Pharmacology, School of Pharmacy, Jeonbuk National, Jeon-Ju, 561-756, Korea

Fax) +82-31-270-4037; Phone) +82-31-270-4038

Comments and Suggestions for Authors

In the revised manuscript and supplementary,

  1. I didn't found 1c  in Figure 1 of supplementary.
  • Author response: Thank you. We added 1c in the supplementary figure and requested the editor to replace the new one with the old one, as below.

  1. In Figure 4A, the bands of JNK and p-JNK are not changed. 
  • Author response: Thank you. I have added figure 4A to the manuscript.

  1. I didn't found 1H  in Figure 1.
  • Author response: Thank you very much. I have added figure 1H to the manuscript.

Round 3

Reviewer 2 Report

The problems have been solved.